# Float-stacked graphene–PMMA laminate

Seung-Il Kim[1,2,3,7], Ji-Yun Moon[1,2,3,7], Seok-Ki Hyeong[1,2,4,7], Soheil Ghods[1,2], Jin-Su Kim[1,2], Jun-Hui Choi[1,2], Dong Seop Park[2], Sukang Bae [4], Sung Ho Cho[5] ✉, Seoung-Ki Lee [6] ✉ & Jae-Hyun Lee [1,2,4] ✉

Semi-infinite single-atom-thick graphene is an ideal reinforcing material that can simultaneously improve the mechanical, electrical, and thermal properties of matrix. Here, we present a float-stacking strategy to accurately align the monolayer graphene reinforcement in polymer matrix. We float graphene-poly(methylmethacrylate) (PMMA) membrane (GPM) at the water–air interface, and wind-up layer-by-layer by roller. During the stacking process, the inherent water meniscus continuously induces web tension of the GPM, suppressing wrinkle and folding generation. Moreover, rolling-up and hot-rolling mill process above the glass transition temperature of PMMA induces conformal contact between each layer. This allows for pre-tension of the composite, maximizing its reinforcing efficiency. The number and spacing of the embedded graphene fillers are precisely controlled. Notably, we accurately align 100 layers of monolayer graphene in a PMMA matrix with the same intervals to achieve a specific strength of about 118.5 MPa g$^{-1}$ cm$^3$, which is higher than that of lightweight Al alloy, and a thermal conductivity of about 4.00 W m$^{-1}$ K$^{-1}$, which is increased by about 2,000 %, compared to the PMMA film.

High-performance, low-power, and small-size semiconductor devices have been intensively studied in the last decades to overcome the physical limitations of the existing system and boost computing efficiency[1–3]. To implement electronic products based on these emerging devices in the near future, various components that can support the devices should be developed together; for example, compact electronic devices require multi-functional materials that are lightweight and flexible but can protect them from external shocks, while quickly dissipating heat generated from the device[4–7]. A composite is a mixture of the structure and characteristics of two or more materials, and in general, the physical and chemical properties of the basic matrix can be enhanced by additive reinforcements[8]. Thus, by adding appropriate reinforcing materials that can satisfy the requirements of each function, multi-functional materials and components can be designed.

Graphene, which is a semi-infinite filler composed only of strong covalent bonds on a plane, has attracted enormous attention as an ideal reinforcing material that, despite its small addition, is capable of simultaneously improving the mechanical, thermal, and electrical characteristics of matrix materials[9–13]. Typical graphene-reinforced nanocomposites were manufactured by dispersing the flake-type graphene fillers (e.g., graphene oxide, reduced graphene oxide, functionalized graphene oxide, etc.) in a polymer matrix[14–16]. However, due to the high defect density and limitation of the lateral size of the graphene flakes, the actual measured physical properties have so far not met the expectations[17]. In particular, as the number of graphene flakes layers approaches the monolayer, both uniform dispersion and alignment control become difficult, due to the strong van der Waal (vdW) interaction[18]. On the other hand, when large-area high-quality graphene ensured through the bottom-up chemical vapor deposition

[1]Department of Energy Systems Research, Ajou University, Suwon 16499, Korea. [2]Department of Materials Science and Engineering, Ajou University, Suwon 16499, Korea. [3]Mechanical Engineering & Materials Science, Washington University in St. Louis, Saint Louis, MO 63105, USA. [4]Functional Composite Materials Research Centre, Institute of Advanced Composite Materials, Korea Institute of Science and Technology, Wanju 55324, Korea. [5]A Development Team, Samsung Display, Asan 31454, Korea. [6]Department of Materials Science and Engineering, Pusan National University, Busan 46241, Korea. [7]These authors contributed equally: Seung-Il Kim, Ji-Yun Moon, Seok-Ki Hyeong. ✉e-mail: sungho19.cho@samsung.com; ifriend@pusan.ac.kr; jaehyunlee@ajou.ac.kr

(CVD) method is used as a filler, the reinforcement effect that is determined by the aspect ratio and alignment of the filler can be maximized[10]. Numerous efforts have been made to increase the graphene number of layers (volume fraction) in the nanocomposite. For example, by repetitive stacking of graphene and poly(-methylmethacrylate) (PMMA) film, 16-layered graphene/PMMA nanocomposites were demonstrated[12]. Nanocomposites of (320 and 1,024) layers were implemented by scrolling and folding the aligned graphene/PC composite, respectively[10,19]. Accordingly, it was confirmed that by adding a strong graphene filler in a high-volume ratio to the softer polymer matrix, high mechanical and electrical improvement can be achieved. However, the process of transference of the graphene and the coating of the matrix materials is a complex and tedious repetitive task. Several approaches (e.g., scrolling and folding) to exponentially increase the volume fraction of the graphene in the matrix have at the same time resulted in a decrease in the in-plane scale. Above all, during the water-assisted stacking and folding process, wetting of the surface of the prepared composite film is inevitable. Numerous studies have verified that water could be spontaneously trapped at the interlayer of the laminates during the wet-transfer process, which leads to the formation of mechanical defects such as bubbles, folding, and wrinkles[20,21]. The deficient interface of the laminate could induce inconsistent fracture strength independent of the graphene volume fraction, limiting the analysis of the strain-to-failure and tensile strength of the composite.

Herein, we presented a scalable manufacturing approach, a float-stacking strategy, for multi-functional graphene-PMMA laminates (GPLs), in which highly crystalline semi-infinite graphene fillers are uniformly aligned in the polymer matrix. To quickly and precisely insert monolayer graphene into the PMMA matrix without mechanical destruction, we floated the delicate and thin graphene−PMMA membrane (GPM) that only relies on water support on a water−air interface and wound it up layer-by-layer. During the stacking process, the inherent water meniscus at the interface between the GPM and deionized (DI) water continuously induced web tension of the GPM, suppressing the structural defects (e.g., wrinkles and folding). Moreover, by the rolling-up and hot-rolling mill process above the glass transition temperature ($T_g$), the viscosity of the highly elastic PMMA film was increased, inducing dry and conformal contact between each GPM layer as well as pre-tension of the GPLs. Through the stress and thermal conducting analysis, we confirmed that both the mechanical and thermal properties of the laminated graphene−PMMA composite were significantly enhanced, and followed the rule of mixtures. When we embedded 100 layers of monolayer graphene in a PMMA matrix with the same intervals (-180 nm), even though the volume fraction of graphene in the composite was only 0.19 vol.%, the specific strength was about 118.5 MPa g$^{-1}$ cm$^3$, which is higher than that of light-weight Al alloy, and the thermal conductivity is about 4.00 W m$^{-1}$ K$^{-1}$, which compared to the PMMA film, was increased by about 2000 %.

## Results

### Manufacture of the GPL

We have three main steps to achieve the GPL, which process accurately stacks large-area monolayer CVD-graphene in the PMMA matrix: (i) Preparation of a thin-PMMA membrane with a high-crystalline monolayer graphene reinforcement, (ii) Float-membrane stacking process capable of conformal laminating a GPM having a thickness of only a few hundreds of nanometers, and (iii) Hot-rolling mill process of the stacked-GPM to induce pre-tension and maximize the interface strength between the graphene and PMMA (Fig. 1(i−iv)).

In detail, we first synthesized high-quality monolayer graphene with a large-domain size on the Cu foil to minimize the mechanical and thermal degradation caused by line defects in graphene[22,23]. The as-prepared graphene has a monolayer area of 99 % or more. and an average grain size of (14.18 ± 3.41) µm (Supplementary Figs. 1 and 2).

We spin-coated a PMMA film on the graphene-grown Cu foil, as a supporting layer and polymer matrix. Because the thickness of the GPM is less than hundreds of nanometers, handling it on a dry floor without physical damage is challenging. Thus, we floated the delicate GPM on a water−air interface after wet-etching of the bottom Cu foil. The position of the float-GPM was finely adjusted without physical damage to make contact with the cylinder-type polytetrafluoroethylene (PTFT) roller. When the edge of the float−GPM touches the PTFT roller, the GPM bonds to the PTFT through hydrophobic interaction. After that, the PTFT roller is lifted onto the water surface and wound up layer-by-layer to stack the GPM.

During the stacking process, the water meniscus, which continuously induces web tension of the GPM, is formed at the interface between the GPM and DI water, due to capillary action (Fig. 1(ii)). The adhesion force caused by the meniscus can be determined by the work of adhesion at the solid (S) − liquid (L) interface ($W_{sl}$): $W_{sl} = \gamma_S + \gamma_L - \gamma_{SL} = \gamma_L(1 + \cos\theta)$, where, θ is the contact angle, and γ is the surface free energy[24,25]. The contact angle of the free-standing GPM is measured to be (42 ± 3)°, and the surface tension value of the DI water is 72.80 mJ m$^{-2}$ [26]. The calculated adhesion energy of the water meniscus formed on the rolling GPM is 126.90 mJ m$^{-2}$, which is higher than that of the stacked GPM (101.45 mJ m$^{-2}$). Therefore, tension is continuously applied to the GPM, suppressing the structural defects (e.g., wrinkles and folding) in the stacking process (See Supplementary Discussion 1 for more details). In addition, we heated the PTFT roller above the glass transition temperature of PMMA ( > 120 °C), to further induce conformal contact between the GPMs[27,28]. After stacking as many layers of the GPM as we desired, we cut and separated it from the PTFT roller, obtaining a free-standing laminated graphene−PMMA composite. Lastly, we nearly eliminated the interlayer voids and bubbles presented at interlayers by squeezing the stacked−GPM through a hot-rolling mill process (the effect and detailed procedures of heat-treatment are discussed in the following text). Figure 1b shows a photo image of the 18.32 ± 0.63 µm thick, 9.8 cm × 4.3 cm GPL, in which 100 layers of GPM are uniformly stacked. Aligned graphene layers within the GPL with equal intervals were confirmed by Raman depth profiling (Supplementary Fig. 3). Cross-sectional scanning electron microscopy (SEM) and transmission electron microscopy (TEM) images of the GPL exhibit a monolayer graphene reinforcement that is placed in between the PMMA matrix (Fig. 1c, d).

**Mechanical properties of the GPLs**. The volume fraction ($V_G$) and the interval of the graphene fillers can be simply adjusted by the concentration of the PMMA and the spin-coating speed (Supplementary Fig. 4). We prepared the GPLs in which the graphene fillers of (0, 10, 25, 50, 75, and 100) layers, referred to as GPL − 0, GPL − 10, GPL − 25, GPL − 50, GPL − 75, and GPL −100, respectively, were aligned (Fig. 2a). For the tensile stress test, the GPL samples were prepared with a length of 3 cm, and they all had similar (or almost the same) thicknesses (Supplementary Fig. 5 and Supplementary Table 1). Figure 2b shows typical stress−strain curves for the GPLs. The average tensile strength of the GPL − 0 and GPL −100 increased by 277.5 % from ((79.60 ± 4.10) to (141.29 ± 3.29)) MPa (Fig. 2c). The tensile strength of the GPL − 10 to GPL − 75 was ((83.99 ± 5.54), (97.99 ± 6.98), (111.11 ± 1.18), and (121.31 ± 6.65)) MPa, respectively. This trend is also observed in the results of the Young's modulus. As the number of embedding graphene filler increased from (0 to 100), The Young's modulus of the GPL was enhanced by 261.26 % from ((3.33 ± 0.15) to (5.37 ± 0.31)) GPa (Fig. 2d). The Young's modulus of GPL − 10 to GPL − 75 was ((3.40 ± 0.17), (3.92 ± 0.22), (4.24 ± 0.10), and (4.70 ± 0.19)) GPa, respectively (see Supplementary Table 2). To clarify the mechanical behavior of GPL, finite element method (FEM) simulations were employed to model graphene-PMMA structures corresponding to different numbers of graphene layers (0, 10, 25, 50, 75, and 100 layers) (Supplementary Fig. 6). An inverse relationship trend has been

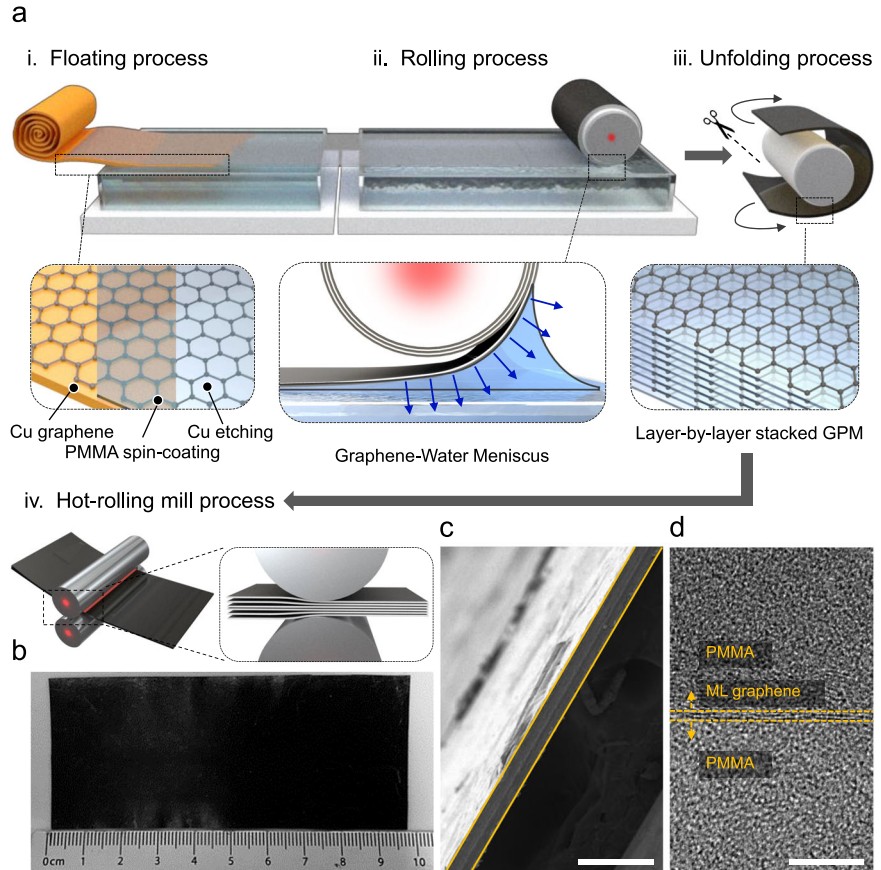

**Fig. 1 | Manufacture of the graphene-PMMA laminate (GPL) via float-stacking process. a** Schematic of the float-stacking process of the GPL. (i) Floating the graphene–PMMA membrane (GPM) on DI water bath after wet-etching of the bottom Cu foil, (ii) Layer-by-layer stacking of GPMs by rolling process, (iii) Cutting and unfolding of stacked-GPM, and (iv) Hot-rolling mill process of the stacked-GPM. **b** Photograph of the as-prepared GPL with 100 layers of GPM (GPL − 100). **c, d** Cross-sectional SEM and TEM images of the GPL − 100. Monolayer graphene is placed between the PMMA matrix without structure defects. Scale bars: 60 μm and 5 nm, respectively.

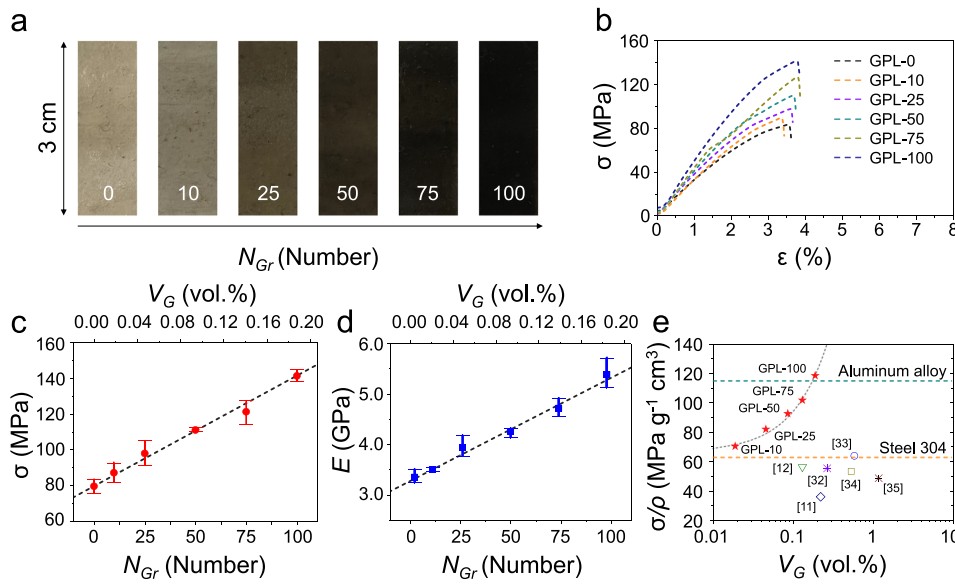

**Fig. 2 | Mechanical characterization of the graphene-PMMA laminates (GPLs).** **a** Photograph of the GPLs with different numbers of graphene reinforcement layers of (0, 10, 25, 50, 75, and 100). **b** Typical stress−strain responses of GPLs. **c, d** Fracture strength, and Young's modulus of the GPLs. Both mechanical properties of GPLs linearly increased according to the number (or the volume fraction) of embedded graphene layers. The error bars represent the standard deviations ($n = 3$). **e** Comparison of the specific strength of GPLs with that of the previously reported graphene/PMMA composites. Source data are provided as a Source Data file.

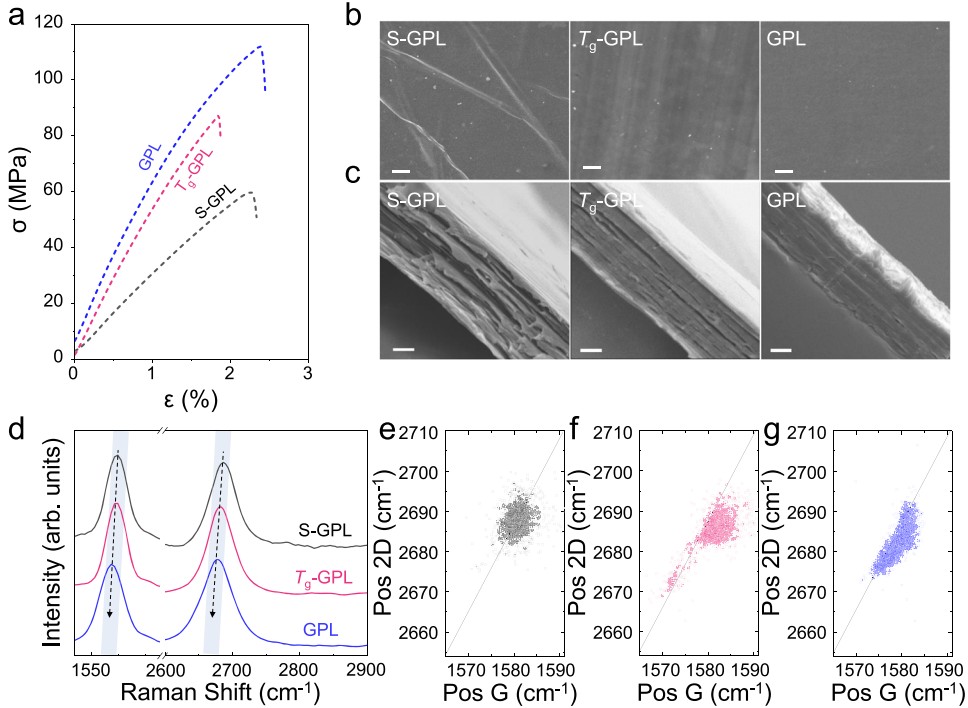

**Fig. 3 | Reinforcing mechanism of the graphene-PMMA laminate (GPL).**
**a** Tensile curve and **b**, **c** SEM images of the simply-stacked GPL (S−GPL), stacked above the $T_g$ ($T_g$-GPL), and GPL, respectively. Scale bar: 50 μm and 1 μm, respectively. **d** Raman spectra of the corresponding GPLs under 532 nm excitation. Both the G and 2D peak positions are gradually red-shifted with the heat-treatment step. **e−g** G peak position vs. 2D peak position of three different GPLs: **e** S−GPL, **f** $T_g$-GPL, and **g** GPL. Source data are provided as a Source Data file.

observed, reveal that as the number of graphene layers increases in the composite, the strain in the structure decreases. Based on simulation results, a reduction in strain at a specific stress level signifies an enhancement in the tensile strength of the structure. Moreover, the experimental stress–strain results agree with the simulation results, verifying the mechanical properties of GPL.

Based on the mechanical characteristics we measured in Fig. 2c, d, the tensile strength ($\sigma_{Gr}$) and modulus ($E_{Gr}$) of the graphene filler were derived by the following rules of mixtures:

$$\sigma_{GPL} = \sigma_{PMMA} \frac{t_{PMMA}}{t_{PMMA} + t_{Gr}} + \sigma_{Gr} \frac{t_{Gr}}{t_{PMMA} + t_{Gr}} \quad (1)$$

$$E_{GPL} = E_{PMMA} \frac{t_{PMMA}}{t_{PMMA} + t_{Gr}} + E_{Gr} \frac{t_{Gr}}{t_{PMMA} + t_{Gr}} \quad (2)$$

where, $\sigma_{GPL}$ is the tensile strength of the GPL, $\sigma_{PMMA}$ is the tensile strength of the PMMA, $E_{GPL}$ is the modulus of the GPL, $E_{PMMA}$ is the modulus of the PMMA, and $t_{PMMA}$ and $t_{Gr}$ are the thickness of PMMA matrix and the graphene reinforcement, respectively. The total thickness ($t_{Gr}$) of the graphene reinforcement was calculated as follows; $t_{Gr} = t_{Mono-gr} \times N_{Gr} \approx 0.34 nm \times N_{Gr}$, where, $t_{Mono-gr}$ is the thickness of the monolayer graphene, and $N_{Gr}$ is the number of graphene fillers in GPL−100. The calculated $\sigma_{Gr}$ and $E_{Gr}$ values are 33 GPa and 1.09 TPa, respectively, which values are close to the values of typical high-quality graphene obtained by the CVD method (A detailed mechanical model of GPL is shown in Supplementary Discussion 2 and Supplementary Fig. 7)[29,30]. We also confirmed that the specific strength of GPL−100 is about 118.5 MPa g⁻¹ cm³, which is higher than that of light-weight Al alloy (Al6061-T6, 115 MPa g⁻¹ cm³) (Fig. 2e)[31]. The specific strength of the previously reported graphene−polymer composites is mostly located at about that of heavy Steel 304, despite the high graphene volume fraction in the light PMMA matrix[11,12,32−35]. The results of these mechanical properties ($\sigma_{Gr}$, $E_{Gr}$, and specific strength) lie within the trend range of the rule of mixtures (dark line in Fig. 2c−e), indicating that the semi-infinite monolayer graphene reinforcement layers are well-aligned in the polymer matrix, and can effectively improve the overall mechanical properties of the composites.

**Reinforcing effect of the well-aligned graphene layers in the GPL**
To explore the remarkable reinforcing factors in our GPL samples, as compared with previous results of graphene/PMMA composites, we prepared three different samples with 25 layers of GPM (A4, 1500 rpm), the simply-stacked GPL (S−GPL), stacked above the $T_g$ ($T_g$-GPL), and GPL, respectively (Supplementary Fig. 8), and then performed a tensile test. The volume fraction of GPL with 25 layers of GPM (A4, 1500 rpm) is 0.190 %. As shown in Fig. 3a, the average tensile strength of the S−GPL was 52.28 MPa, whereas that of the stacked above the $T_g$-GPL was 83.11 MPa, and that of the GPL was 103.72 MPa, which improved the mechanical properties as heat-treatment was conducted (see Supplementary Fig. 9 and Supplementary Table 3).

Although the tension generated by the meniscus at the interface between the GPM and water helps to reduce structural defects during the float-stacking process, physically conformal contact between the graphene and PMMA could not be possible, because of the high elastic modulus of PMMA at room temperature (RT). Thus, the wrinkles and hollow regions that lead to interlayer sliding were frequently observed on the surface and interlayer of the S−GPL, respectively[36]. However, rolling above the $T_g$ temperature increased the viscosity of the highly elastic PMMA film, inducing conformal contact between each layer (Supplementary Fig. 10)[37]. Thereafter, the small voids present in the $T_g$-GPL were effectively removed through the hot-rolling mill process (Fig. 3b, c and Supplementary Fig. 11). Note that the transmittance in the visible light (550 nm) region of the S−GPL and GPL with five sheets of the graphene embedded therein

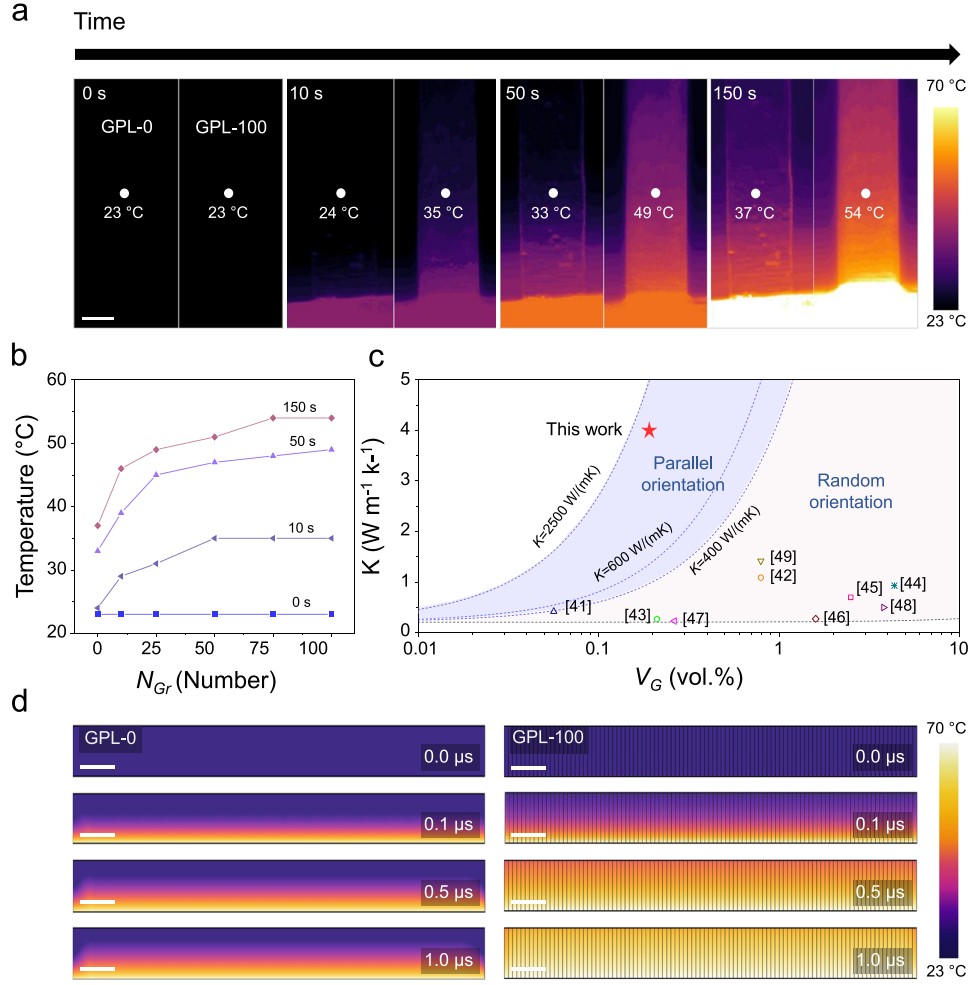

**Fig. 4 | Thermal properties of the graphene-PMMA laminates (GPLs). a** Time-dependent IR image from GPL – 0 to GPL – 100. Scale bar: 1 mm. **b** Temperature difference from GPL – 0 to GPL – 100 according to the distance measured from the IR images. The temperature of GPL – 100 was higher than that of GPL – 0 in all sections of the composite. **c** Comparison of thermal conductivity with GPL and various graphene/PMMA composites from the literature against the graphene volume fraction. The blue dashed line calculated through the graphene thermal conductivity ($K = (400$ to $2500)$ W m$^{-1}$ K$^{-1}$) limits the range of parallel orientation, and the random orientation area calculated by the Maxwell–Eucken (ME) model covers various reported graphene/PMMA composites. **d** In-plane thermal simulation of the bare PMMA (GPL – 0) and graphene/PMMA (GPL – 100) composite, with different heat conduction times ($t = (0$ to $1)$ μs). Scale bar: 1 μm. Source data are provided as a Source Data file.

increases from (72.13 to 89.17) %, respectively, indicating a reduction in scattering elements (e.g., wrinkles, voids, and folds) (Supplementary Fig. 12).

Through the strain analysis of the graphene filler, we also confirmed that the pre-tension was applied to the GPL during the heating and hot-rolling mill process. In all samples, the G peak and 2D peaks, which are the Raman signature of graphene, clearly appeared, and as conformal contact was made, both peaks gradually red-shifted (Fig. 3d). To verify the origin of peak shift, we plotted the position of the G and 2D peaks for all spectra obtained from the Raman mapping results of each sample (Fig. 3e–g and Supplementary Fig. 13). The data points of the simply-stacked GPL are evenly distributed from the origin (G peak (1580 cm$^{-1}$) and 2D peak (2687 cm$^{-1}$)). In contrast, the data points acquired from the $T_g$–GPL and GPL were scattered along a line with a slope of 2.2, indicating a uniaxial strain existed in the heat-treated GPLs. Thus, we concluded that through the overall heat-treatment process, the conformal contact of the PMMA matrix and the graphene filler could be realized, and the pre-tension of the composite could be applied, thereby maximizing the reinforcing efficiency (see Supplementary Discussion 3 for more details.).

## Thermal properties of GPL

Single-atom-thick graphene has a high intrinsic thermal conductivity of (600 – 5,300) W m$^{-1}$ K$^{-1}$, which is contributed by the phonon components as a result of the strong in-plane bonding of the carbon atoms[38–40]. Thus, even when used in a small amount as a heat transfer reinforcement, we believe that the thermal dissipation performance of the graphene-embedded composite could be effectively enhanced. Real-time visualization of the heat conduction of the GPL samples was measured with an IR camera (Fig. 4a, see "Methods"). We observed heat transfer from the heat source to the GPLs over time from (0 to 150) s; at the position the same distance away from the heater (1.5 mm), the temperature difference between GPL – 0 and GPL – 100 gradually increases over time (Fig. 4b and Supplementary Fig. 14).

Figure 4c shows the thermal conductivity of the GPLs and the previously reported graphene–PMMA composites[41–49]. The results could be divided into two different areas, depending on the alignment of the graphene filler in the composite (see Supplementary Discussion 4 for more details). The thermal conductivity results of previous works based on the nanoplatelet (flakes) types of graphene and graphene oxide are mostly distributed within a random orientation regime that requires a high graphene volume fraction to improve the

thermal properties of composites. In contrast, the thermal conductivity results of our GPL samples are plotted in a parallel orientation regime. The thermal conductivity of GPL−100 we measured here is about 4.00 W m$^{-1}$ K$^{-1}$, even though the volume fraction of graphene in the composite is only 0.19 vol.%, which compared to the PMMA film, is increased by about 2,000 %. Note that the extracted thermal conductivity of graphene filler in GPL is 1992.28 W m$^{-1}$ K$^{-1}$, indicating that the high-quality CVD-graphene maintained its superior thermal conducting property during the overall manufacturing process.

To understand the thermal conduction mechanism of the well-aligned graphene fillers in the GPLs, we modeled an 18 μm thick graphene−PMMA composite in which different numbers of graphene layers of (0, 10, 20, 50, and 100) were embedded (Supplementary Fig. 15). The heat diffusion of the amorphous PMMA is much slower than that of conducting crystals, because of the disordered vibrations and rotations of the polymer atoms ($k_{PMMA} = (0.19 - 0.22)$ W m$^{-1}$ K$^{-1}$). In the case of the GPL in which PMMA and high-quality graphene are sequentially stacked, the semi-infinite graphene filler acts as a shortcut to heat transport. Therefore, heat transfer to PMMA can be induced both by the heater and by the embedded graphene filler; consequently, the temperature of the GPL is uniformly and quickly increased over time (Fig. 4d). As we expected, the thermal diffusion of the composite is proportional to the number of graphene layers (Supplementary Fig. 16). Thus, we believe that the interface between graphene and PMMA in GPL is clean and sharp, without structural defects that might interfere with the heat transport.

In summary, we manufactured multi-functional composites with precisely controlled layers and spacing of the semi-infinite graphene reinforcement in the polymer matrix. By rolling-up the floating GPM on DI water, we were able to quickly and accurately laminate the GPMs. Continuously induced web tension caused by capillary actions and a series of heat treatments effectively eliminated the structural defects presented at the interlayers (voids, wrinkles, and folds), maximizing the reinforcing efficiency of the graphene filler. As a result, we observed that the mechanical and thermal properties of GPL were significantly enhanced, even when the volume fraction of graphene was just 0.19 %. We believe that our float-stacking strategy will be of great help in the realization of functional nanocomposite based on low-dimensional nanomaterials, which are ideal semi-infinite reinforcements that are difficult to disperse.

## Methods

### Preparation of the GPM

Monolayer graphene was grown on 15 cm × 15 cm Cu foil (25 μm thick, 99.9 %, Basic Copper) with a modified CVD system. For continuous rolling of the GPM, we synthesized graphene sheets as large as 15 cm in length, but the temperature gradient in the furnace can sometimes cause inhomogeneous graphene growth[50]. The Cu foil wrapped with a 2-inch quartz tube is loaded in the center of the 3-inch quartz tube, minimizing the temperature gradient, and ensuring the crystallinity of the graphene. The furnace was ramped to 1,000 °C for 30 min with 100 sccm H$_2$ gas flow with the pressure of 100 torr. For large grain size, the temperature and gas flow were maintained for 30 min after reaching 1000 °C, to anneal the Cu foil. After that, 10 sccm of CH$_4$ gas was flowed and maintained at 90 mtorr for 1 h. After the growth, the CH$_4$ gas flow was stopped, and the graphene/Cu cooled rapidly to RT. Then, PMMA (950k, Microchem) dissolved in anisole was coated on the as-grown graphene, and satisfied sufficient thickness for floating with different concentrations of PMMA solution (4, 8, and 12 wt.%) and spinning speeds (500, 1000, 1500, 2500, and 4000) rpm to modulate the graphene volume fraction (Supporting information Fig. 4). The PMMA−graphene/Cu sheet was divided to the width of the roll, and floated on Cu etchant (ammonium persulfate). The Float−GPM was then transferred, and rinsed with deionized water 3 times for 10 min each.

### Manufacture of the GPL

The stacking of GPM is manufactured by the rolling-up process. In detail, GPM floating on the water moves along the rails on both sides and is precisely positioned. After the first GPM is rolled on the PTFT, the GPM is sequentially stacked by rotation of the semi-automatic roller. The roller rotates backward gently ( < 0.1 rpm) to completely evaporate the moisture on the surface of the GPM. Since the subsequent GPM is connected to the previous membrane, the layer-by-layer stacking proceeds continuously. The water meniscus acts as a bridge that stably connects the GPM with a roller, and also induces tension in the roll. Meniscus web tension enables high-quality rolling, without additional roll for tension control. For the $T_g$−GPL, the cylindrical heater is connected inside the PTFT roller, and conducts the temperature above the PMMA $T_g$ evenly throughout the stacked layer. After stacking is completed, heating is maintained for 5 min, to remove residual moisture. The free-standing GPL separated from PTFT is passed through heated rolling mills. In order to measure the pressure of the hot rollers during the hot-rolling mill process, we placed a piezo pressure sensor (MD 30-60) between the hot rollers. We confirmed a force of 78.48 N, corresponding to a compressive pressure of 8.35 MPa, was applied to the $T_g$-GPL.

### Characterization

Surface scan and profile measurement with Alpha-step (P-500, KLA Tencor) were performed on single layers of GPM. The measured film was subjected to PMMA spin-coating of the as-grown graphene and copper etching process as in the stacked GPM and transferred onto a 300 nm SiO$_2$/Si substrate. Raman spectroscopy (WITEC Raman system) with an excitation energy of 2.33 eV (532 nm) was performed for graphene quality evaluation and GPL mapping. Raman mapping was carried out on an area of 35 μm x 35 μm in 0.5 μm steps. To avoid laser-induced heating of the polymer and graphene, the laser power was less than 1 mW. Depth profiling was performed by Renishaw inVia™ confocal Raman microscopy (-10 μm). High-resolution SEM (JSM-IT500HR, JEOL) was used to observe the cross-sectional and surface morphology of the GPL. For the cross-sectional TEM images, the GPLs were cut using an ultramicrotome (EM UC7, Leica). TEM images were obtained by HR − TEM (Libra 200 HT MC, Carl Zeiss) operating at 200 kV, for visualization of the aligned monolayer graphene. Transmittance and reflectance were measured using UV−Vis−NiR spectrophotometry (Cary 5000, Agilent). Specific heat of LGC was performed by DSC (DSC 204 F1 Phoenix, NETZSCH).

### Mechanical analysis

Mechanical properties were measured using a FAVIMAT+ (Textechno, GmbH) with 210 cN load cell. Samples were cut with a razor blade to a total length of 15 mm and a width of 1 mm. The thickness of each specimen was determined as the average of repeated thickness measurements of the corresponding composites. For precise testing with control of the gauge length and grip area, the specimen was mounted on a paper frame. The specific gauge length of the specimen was 10 mm, and the tensile test was performed at a displacement rate of 0.08 mm/s (strain rate of 0.008 /s). From the recorded stress−strain values, Young's modulus was estimated in the region with a linear slope in the initial strain (<0.5 %). The average result of Young's modulus was obtained through five samples.

### FEM simulation

The strengths of distinct structures with varying numbers of graphene layers are evaluated through FEM analysis employing COMSOL Multiphysics. For uniaxial tension testing of graphene-PMMA structure, we directly modeled the geometry corresponding to GPL-0 to 100 and employed the linear elastic relationships provided in the mechanics section for accurate estimation. In this simulation, a fixed boundary was established, and an applied force was introduced to the free

boundary, and the resulting deformation, quantified as static displacement, was measured. Tensile simulation was computed according to the following relation as $-E \nabla \cdot u = F_v/A$, where $E$ represents the elastic constant, $A$ denotes the force application area ($W \times t$: 0.6 mm × 18 μm), $u$ represents displacement, $F_v$ represents the applied force inducing strain.

## Thermal analysis

The temperature profile was measured by thermal emission microscopy, Thermos mini C10614-02 (Hamamatsu), while applying the current with a Keithley 2440 (Tektronix) source meter. Thermal conductivity was calculated by measuring the thermal diffusivity, density, and heat capacity. The thermal diffusivity was measured using a scanning laser heating thermal diffusivity meter (Laser PIT, ULVAC-Riko, Inc.). The specimen was attached with silver paste to the thermocouple wire of the sample holder. After the sample was placed in a vacuum chamber, the in-plane thermal diffusivity was measured while irradiating at a period of 1/s using a 685 nm diode laser.

## Thermal simulation

The thermal conductivity of the device was simulated using a 2D structure of monolayer graphene and PMMA on the heat transfer module of COMSOL Multiphysics (V5.5), in which the thermal conductivity calculation based on convective heat flux equations was carried out. This simulation was performed for different numbers of graphene layers (0 to 100), and the simulations display the key effect of graphene in increasing the thermal conductivity of the structure. The thickness of the graphene and PMMA were set as 0.34 nm and 18 μm, respectively.

## Reporting summary

Further information on research design is available in the Nature Portfolio Reporting Summary linked to this article.

# Data availability

Source data are provided in this paper. All other data that support the plots within this paper and other findings of this study are available from the Supplementary Information or the corresponding authors upon request. Source data are provided in this paper.

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

## Acknowledgements
This work was supported by a National Research Foundation (NRF) of Korea grant (NRF-2021R1A2C2012649).

## Author contributions
S.-I.K., J.-Y.M. and J.-H.L. conceived the research. S.-I.K., J.-S.K., J.-H.C. and J.-Y.M. performed the float-stacked GPL, designed experiments, and analyzed the data. S.-K.H., S.B. and S.-K.L. measured tensile tests, and conducted thermal emission microscope and data analyses. S.G. conducted the thermal simulation of the 2D structure and carried out convective heat flux equations for thermal conductivity calculation. D.S.P. and S.H.C. participated in preparing GPM and data analyses. S.-I.K., J.-Y.M. and J.-H.L. wrote the original draft, while S.-I.K., S.-K.H., S.B., S.-K.L. and J.-H.L. reviewed & edited the manuscript. S.-K.L. and J.-H.L. supervised the research. J.-H.L. conceived the idea for the entire project. All authors discussed the results and commented on the manuscript.

## Competing interests
The authors declare no competing interests.
