## [Peer Review File · Nature Communications]

REVIEWER COMMENTS

Reviewer #1 (Remarks to the Author):

This work fabricated a multi-functional graphene–PMMA laminate (GPL) with monolayer graphene flakes aligned in the matrix. The authors demonstrated that the GPLs show both the mechanical and thermal properties were significantly enhanced, and followed the rule of mixtures. These performance enhancements are significant. However, the design concept is not new, which is similar to the studies of refs. 12 and 10, especially ref. 12. After carefully read these papers and found only the preparation method is a new one. Thus, the fabrication of GPLs should be the highlighted point of this work, but the scalable fabrication of GPLs using this method exists a big problem for application, owing to their lengths or widths depends on the diameter of roller which is limited. Comparing to the other similar graphene-based laminates in ref. 12 and (NAT. COMMUN. (2021) 12:4655), the advantage of this design is not noticeable.

Some other issues also should be addressed in this work. Detailed comments are listed below.

1. It seems that ref. 12 is an important reference related to this paper. The laminate structure and material components are almost identical to the composites in this ref. Also, the mechanical and thermal properties were studied in ref. 12. Thus, the authors should highlight the difference with this ref. in the manuscript.
2. It is well known that the final thickness of the GPLs depends on the hot rolling pressure, but the relationship between the thickness and pressure is not demonstrated. Besides, how does the author control the thickness of the GPLs? This is important for readers to reproduce the methods.
3. The author roughly concluded that the voids and bubbles were completely eliminated in this study. This statement lacks experimental supports. The reviewer thinks this expression “completely eliminated” is overstated and not proper.
4. Why the thickness slightly decreased for GPLs (GPL-25 to GPL-100) with high graphene filler contents? There needs a proper explanation. Besides, how to obtain the similar thicknesses during the manufacturing process?
5. The mechanical properties in Fig. 2 were measured through using samples with similar total thicknesses. This means that the thickness of embedded PMMA layers in GPLs is different in these experiments. How about the mechanical properties of GPLs with same PMMA layer thickness but different total thickness?
6. The authors use the number of layers to characterize the mechanical property. However, the volume content of graphene is the direct factor for the mechanical performance. The volume content of graphene should be given in the Fig. 2c and d.
7. The authors analyze the mechanical properties of S-GPL, Tg-GPL, and GPL, and indicated that the mechanical performance of GPL was largely enhanced. The reviewer thinks this enhancement mainly caused by the hot-rolling mill process which resulting in a compact interface between PMMA and

graphene; the reinforcement efficiency of float stacking process may have a positive effect but not the main factor. My question is how about the mechanical performance of multilayered graphene–PMMA laminate fabricated by simply stacking and hot-pressing. The simple and easy manufacturing method is significant for engineering practices.

Reviewer #2 (Remarks to the Author):

This paper describes a new method that uses the free surface of a liquid as a support for the production of composites alternating sheets of PMMA with sheets of polycrystalline graphene produced by CVD. The topic is innovative and treated with the necessary experimental scientific rigor. There are also two mechanical and thermal analytical analyses and one thermal numerical analysis. The strength of the paper is certainly the experimental part. However, the mechanical analytical approach is important as it suggests how the composite is done properly, returning a Young's modulus equal to the theoretical graphene modulus of about 1 TPa and a strength of about 1/3 of the theoretical value of 100GPa but compatible with the expected values of polycrystalline graphene obtained via CVD. I would add in the main text in the mechanics section the thicknesses of PMMA and/or the whole specimen i.e. all the data entered in the rule of mixture. I would avoid writing that capillarity forces are able to remove all defects as initially reported in the paper as unlikely, and as moreover admitted in the second part of the paper. The length of the specimens is 1 cm and could lead to inherently higher strength measurements given the smaller size (and relative lower likelihood of finding defects) than the traditional ones at which composites are tested (closer to 10 cm than 1). I think the paper is much improvable in the analytical and numerical parts, but since it is primarily experimental it can be published nearly in this form in Nature Communications. Almost every time I see Pa writted with p and it should be corrected. (Using a law derived from atomistic simulations from the grain size reported in the paper I find a graphene resistance of about 80 and not 33 GPa as calculated in this paper, but there could be many reasons for this difference).

Authors' Reply to the Reviewers' Comments on:

Float-stacked multi-functional graphene–PMMA laminate

By Seung-Il Kim, Ji-Yun Moon, Seok-Ki Hyeong, Soheil Ghods, Jin-Su Kim, Jun-Hui Choi, Dong Seop Park, Sukang Bae, Sung Ho Cho, Seoung-Ki Lee, and Jae-Hyun Lee

We deeply appreciate the effort that the editor and reviewers have taken in reviewing our manuscript. Here we include the answer to the referees and the revised manuscript. Changes have been carried out according to the comments and highlighted as red color in the revised version of the manuscript. We hope that our reply properly addressed the points of the comments.

I. Response to comments of Reviewer #1

This work fabricated a multi-functional graphene–PMMA laminate (GPL) with monolayer graphene flakes aligned in the matrix. The authors demonstrated that the GPLs show both the mechanical and thermal properties were significantly enhanced, and followed the rule of mixtures. These performance enhancements are significant. However, the design concept is not new, which is similar to the studies of refs. 12 and 10, especially ref. 12. After carefully read these papers and found only the preparation method is a new one. Thus, the fabrication of GPLs should be the highlighted point of this work, but the scalable fabrication of GPLs using this method exists a big problem for application, owing to their lengths or widths depends on the diameter of roller which is limited. Comparing to the other similar graphene-based laminates in ref. 12 and (NAT. COMMUN. (2021) 12:4655), the advantage of this design is not noticeable. Some other issues also should be addressed in this work. Detailed comments are listed below.

Q1. It seems that ref. 12 is an important reference related to this paper. The laminate structure and material components are almost identical to the composites in this ref. Also, the mechanical and thermal properties were studied in ref. 12. Thus, the authors should highlight the difference with this ref. in the manuscript.

→ Thank you for the reviewer's helpful comments. As the reviewer suggested, we carefully studied the ref. 12 and its related references. As the reviewer mentioned, CVD-graphene reinforced laminate structure with the polymer (or metal) matrix have been demonstrated by several groups. They confirmed that the physical properties of the matrix can be dramatically enhanced through the addition of a trace amount of the aligned graphene reinforcement.

In many cases, they have employed a repetitive wet-transferring and stacking strategy to manufacture laminate structures. For examples, Kim et al. produced a multi-stacked metal (Cu or Ni)-graphene laminate by repetitive metal deposition and wet-transfer process [Ref. *Nat. Commun.* **4**, 2114 (2013)]. Ref. 11 and 12 reported on the fabrication of laminates with PMMA-matrix and graphene-reinforcement, by a repetitive wet-transfer process [Ref. *ACS Appl. Mater. Interfaces* **7**, 10702-10709 (2015); *Nat. Commun.* **12**, 1-9 (2021)]. Prof. Strano's group has reported novel experimental method for layered, moving beyond simple stacking approach [Ref. *Science* **353**, 364-367 (2016)]. They demonstrated up to 320 graphene layer incorporated fiber nanocomposite by simple scrolling of graphene/polymer films. Prof. Ruoff's group has reported folded laminate structure via a water-assisted folding process on patterned substrates [Ref. *Adv. Mater.* **30**, 1707449 (2018)].

→ We agreed that the wet-transfer process is a reliable and straightforward fabrication method for CVD-graphene based laminates. However, during the water-assisted stacking and folding process, wetting of the surface of the prepared film (matrix-graphene) is inevitable. Numerous studies have verified that water could be spontaneously trapped at the interlayer of the laminates during wet-transfer process, which leads to the formation of mechanical defects such as bubbles, folding, and wrinkles [Ref. *Adv. Funct. Mater.* **26**, 2070 (2016); *Proc. R. Soc. Lond. A Math. Phys. Sci.* **324**, 301–313 (1971)]. The deficient interface of the laminate could induce inconsistent fracture strength independent of the graphene volume fraction, limiting the strain-to-failure and tensile strength of the composite. Although the removal of trapped water is attempted through gentle heat treatment in a vacuum for several hours, the reinforcing efficiency is still low, indicating that the defects could still exist at the interlayer of laminates. Above all, it takes a long and effort to insert the semi-infinite graphene with regular intervals.

→ On the contrary, we have stacked dry-side of the graphene-polymer membrane (GPM) floated on DI water (not in DI water), enabling us to avoid the adversary effects of moisture between the interfaces. Moreover, the inherent water meniscus at the interface between the GPM and DI water continuously induced web tension of the GPM, suppressing the structural defects (e.g., wrinkles and folds) during the stacking process. As a result, although the volume fraction of graphene in the composite is only 0.19 vol. % (100 layers of monolayer graphene inserted into the GPL), we achieved the specific strength is about 118.5 MPa/(g/cm³), which is higher than that of lightweight Al alloy, and the thermal conductivity is about 4.00 W/mK, which is increased by about 2,000% compared to the PMMA film.

→ We carefully investigated previously published studies on the graphene-based composite once again. We summarize the mechanical performance of graphene-based composites reported so far

in Table R1. We confirmed that our GPL has the highest reinforcement efficiency at the given graphene volume fraction (0.19%) compared to the previous studies.

Table R1. Reinforcement efficiency of various graphene-based nanocomposite. The modulus and strength of nanocomposites were calculated at the graphene volume fraction of 0.19 vol.%.

Materials	Structure	Experimental method	E_c (V_g : 0.19%) (GPa)	σ_c (V_g : 0.19%) (MPa)	Ref
Graphene/PMMA	Laminate	Float-stack	5.37	141.29	This work
Graphene/PMMA	Laminate	Wet-transfer	3.68	73.80	ACS Appl. Mater. Interfaces 7 , 10702-10709 (2015)
Graphene/PMMA	Laminate	Wet-transfer	3.35	51.59	Nat. Commun. 12 , 1-9 (2021)
Graphene/PC	Laminate (scroll)	Wet-transfer and scrolling	1.27	97.10	Science 353 , 364-367 (2016)
Graphene/PMMA	Dispersed	In situ polymerization	4.34	56.09	Mater. Chem. Phys. 136 , 43-50 (2012)
bwGO/PMMA	Dispersed	Melt mixing	2.36	67.76	Compos. Sci. Technol. 88 , 158-164 (2013).
CRGO/PMMA	Dispersed	Emulsion polymerization	3.58	58.22	Compos. Sci. Technol. 134 , 144-152 (2016).
FLG/PMMA	Dispersed	Melt mixing	2.07	58.41	Compos. Sci. Technol. 111 , 17-22 (2015).

→ We believe that our float-stacking strategy not only effectively eliminated the defects (structural defects and water trap) but also induced pre-tension, ensuring maximization of the reinforcing efficiency of the CVD-graphene filler. To further emphasize the novelty of our results, we revised our manuscript as follow.

→ In page 3-4,

... Several approaches (e.g., scrolling and folding) to exponentially increase the volume fraction of the graphene in the matrix have at the same time resulted in a decrease in the in-plane scale. Above all, during the water-assisted stacking and folding process, wetting of the surface of the prepared composite film is inevitable. Numerous studies have verified that water could be spontaneously trapped at the interlayer of the laminates during wet-transfer process, which leads to the formation of mechanical defects such as bubbles, folds, and wrinkles^{20,21}. The deficient interface of the laminate could induce inconsistent fracture strength independent of the graphene volume fraction, limiting the analysis of the intrinsic strain-to-failure and tensile strength of the composite.

... Moreover, by the rolling-up and hot-rolling mill process above the glass transition temperature (T_g), the viscosity of the highly elastic PMMA film was increased, inducing dry and conformal contact between each GPM layer.

[20] *Adv. Funct. Mater.* **26**, 2070–2077 (2016). [21] *Proc. R. Soc. Lond. A. Math. Phys. Sci.* **324**, 301–313 (1971).

Q2. It is well known that the final thickness of the GPLs depends on the hot rolling pressure, but the relationship between the thickness and pressure is not demonstrated. Besides, how does the author control the thickness of the GPLs? This is important for readers to reproduce the methods.

→ We appreciate the reviewer's helpful and relevant comments. The answers to the two comments are as follows.

(1) We placed a piezo pressure sensor at the hot-rollers and measured how much pressure was applied to the GPL. We confirmed that a hot rolling mills with force of 78.48 N, corresponding to a compressive pressure of 8.35 MPa, was applied to the T_g -GPL.

We revised manuscript and added the detailed information of the pressures analysis in the experimental section.

→ In page 11,

... However, by rolling up above the T_g temperature with a gentle pressure of 8.35 MPa applied, the viscosity of the highly elastic PMMA film was increased, inducing conformal contact between each layer (Supplementary Fig. 9).

→ In page 16, Experimental Section,

... In order to measure the pressure of the hot rollers during the hot-rolling mill process, we placed a piezo pressure sensor (MD 30-60) between the hot rollers. We confirmed a force of 78.48 N, corresponding to a compressive pressure of 8.35 MPa, was applied to the T_g -GPL.

(2) The GPLs, we manufactured here, have almost the same thickness because the thickness of the GPL is determined by the thickness of the 1-layer of GPMs. We adjusted the thickness of the GPM by PMMA concentration and spin coating speed. For example, in the case of GPL-10, GPM coated with thick PMMA (PMMA A12, 1500 rpm) was laminated, and in the case of GPL-100, GPM coated with thin PMMA (PMMA A4, 1500 rpm) was laminated, respectively.

→ In order to clarify it, we revised Supplementary Fig. 4. And we summarized average thickness of GPLs and its PMMA coating conditions in Supplementary Fig. 5 and Table 1.

→ In page 8,

... The volume fraction (V_G) and the interval of the graphene fillers can be simply adjusted by the concentration of the PMMA and the spin coating speed (Supplementary Fig. 4). ...

For the tensile stress test, the GPL samples were prepared with a length of 3 cm, and they all had similar (or almost the same) thicknesses (Supplementary Fig. 5 and Supplementary Table 1).

→ In page 15,

... Then, PMMA (950k, Microchem) dissolved in anisole was coated on the as-grown graphene, and satisfied sufficient thickness for floating with different concentrations of PMMA solution (4, 8, and 12 wt.%) and spinning speeds (500, 1,000, 1,500, 2,500, and 4,000) rpm to modulate the graphene volume fraction (Supporting information Fig. 4). ...

→ In Supplementary Information,

Supplementary Fig. 4 | Thickness analysis of the 1 layer (L)-GPM according to the concentration of PMMA solution and coating speed. After spin coating the PMMA on the graphene, the GPMs transferred onto the 300 nm SiO₂/Si. Thickness of the GPMs was measured by the Alpha-step. To manufacture the GPLs of the same thickness, we used different GPMs under the following conditions: GPL-10 (PMMA A12, 1500 rpm), GPL-25 (PMMA A8, 1500 rpm), GPL-50 (PMMA A4, 500 rpm), GPL-75 (PMMA A4, 1000 rpm), GPL-100 (PMMA A4, 1500 rpm). The graphene volume fractions for GPL-10,

GPL-25, GPL-50, GPL-75, and GPL-100 are 0.019 vol.%, 0.046 vol.%, 0.098 vol.%, 0.142 vol.%, and 0.190 vol.%, respectively.

Supplementary Fig. 5 | Representative cross-sectional SEM images of GPLs. Average thickness of GPL-0 to GPL 100 was ((18.64 ± 0.70), (18.35 ± 0.85), (18.61 ± 0.87), (18.72 ± 0.84), (18.95 ± 0.72), and (18.32 ± 0.63)) μm , respectively. Scale bar is 50 μm .

Supplementary Table 1. Statistical thickness analysis of GPLs with different number of graphene layers.

		Thickness (μm)				Thickness (μm)				Thickness (μm)	
GPL-0 (0.0 vol.%)	#1	17.68	GPL-10 (0.019 vol.%)	#1	18.04	GPL-25 (0.046 vol.%)	#1	18.67			
	#2	18.13		#2	19.65		#2	17.07			
	#3	18.59		#3	17.41		#3	19.71			
	#4	19.53		#4	19.14		#4	18.98			
	#5	18.62		#5	17.93		#5	18.44			
	#6	19.31		#6	17.92		#6	18.80			
	Average	18.64 ± 0.70		Average	18.35 ± 0.85		Average	18.61 ± 0.87			
GPL-50 (0.098 vol.%)	#1	19.89	GPL-75 (0.142 vol.%)	#1	19.59	GPL-100 (0.190 vol.%)	#1	18.08			
	#2	19.10		#2	18.38		#2	19.36			
	#3	17.35		#3	19.57		#3	17.59			
	#4	18.31		#4	19.52		#4	17.89			
	#5	18.91		#5	18.71		#5	18.62			
	#6	18.77		#6	17.91		#6	18.44			
	Average	18.72 ± 0.84		Average	18.95 ± 0.72		Average	18.32 ± 0.63			

Q3. The author roughly concluded that the voids and bubbles were completely eliminated in this

study. This statement lacks experimental supports. The reviewer thinks this expression “completely eliminated” is overstated and not proper.

→ We appreciate the reviewer's critical and relevant comments.

We further performed optical microscope analysis (both bright field (BF) and dark field (DF) mode), and AFM analysis to precisely verify the morphology of the GPL sample. Note that wet-transferred graphene-PMMA laminate (WT-GPL) was prepared for comparison. Both samples (GPL and WT-GPL) were stacked with 10 layers of GPM (PMMA A4, 1500 rpm). In contrast to the WT-GPL, where various defects (wrinkles, bubbles, holes, and water traps) were observed, no physical defects were found on the surface and interlayers of GPL (Supplementary Fig. 10). The root mean square (RMS) roughness of the GPL was 69.26 nm, which is nearly half that of the WT-GPL.

We believe that these experimental results strongly supported that the float-stacking process could effectively eliminate interlayer defects, allowing us strong interfacial adhesion of laminate. We added the OM and AFM data to the Supplementary Information.

→ In Supplementary Information,

Supplementary Fig. 10 | Morphology analysis of GPL and wet-transferred graphene-PMMA laminate (WT-GPL) with 10 layers of GPM (PMMA A4, 1500 rpm). High magnification (Left) bright field and (Right) dark field OM images of GPL and WT-GPL. In the corresponding AFM images, the root mean square of GPL is 69.26 nm and WT-GPL is 138.40 nm. Scale bar is 50 and 20 μm , respectively.

→ We agree with the reviewer's concern about our expression could be overstated. Thus, we revised the manuscript as follows.

→ In page 7,

... Lastly, we **nearly** eliminated the interlayer voids and bubbles presented at interlayers by squeezing the stacked-GPM through a hot-rolling mill process (the effect and detailed procedures of heat-treatment are discussed in the following text). ...

→ In page 11,

However, by rolling up above the T_g temperature, the viscosity of the highly elastic PMMA film was increased, inducing conformal contact between each layer (Supplementary Fig. 9). Thereafter, the small voids present in the T_g -GPL were **effectively** removed through the hot-rolling mill process (Figs. 3b, c and Supplementary Fig. 10).

Q4. Why the thickness slightly decreased for GPLs (GPL-25 to GPL-100) with high graphene filler contents? There needs a proper explanation. Besides, how to obtain the similar thicknesses during the manufacturing process?

→ We appreciate the reviewer's critical comment. We sincerely apologize that putting only representative cross-sectional SEM images in the manuscript may lead to a misunderstanding that the thickness of GPLs has decreased as the contents of graphene filler have increased. Statistically, there is no significant difference in the thickness of GPLs.

The answers to the two comments are as follows.

(1) In order to obtain statistical data for the thickness of GPLs, we manufactured additional GPLs and measured the thickness of it. The average thicknesses of GPL-0, GPL-10, GPL-25, GPL-50, GPL-75, and GPL-100 were found to be $18.64 \pm 0.70 \mu\text{m}$, $18.35 \pm 0.85 \mu\text{m}$, $18.61 \pm 0.87 \mu\text{m}$, $18.72 \pm 0.84 \mu\text{m}$, $18.95 \pm 0.72 \mu\text{m}$ and $18.32 \pm 0.63 \mu\text{m}$, respectively (See the Supplementary Fig. 5 and Table 1). Graphene filler contents have a maximum volume fraction of 0.19% in GPL-100 (100-layer, 34 nm), which is negligible compared to the total thickness.

We added average thickness data of GPLs in Supplementary Fig. 5 and Table 1, and revised the manuscript as follows.

→ In page 8,

... For the tensile stress test, GPL samples were prepared with a length of 3 cm, and they all had similar (or almost the same) thicknesses (Supplementary Fig. 5 and Supplementary Table 1). ...

→ In Supplementary Information,

Supplementary Fig. 5 | Representative cross-sectional SEM images of GPLs. Average thickness of GPL-0 to GPL 100 was ((18.64 ± 0.70), (18.35 ± 0.85), (18.61 ± 0.87), (18.72 ± 0.84), (18.95 ± 0.72), and (18.32 ± 0.63)) μm, respectively. Scale bar is 50 μm.

Supplementary Table 1. Statistical thickness analysis of GPLs with different number of graphene layers.

GPL-0			GPL-10			GPL-25		
(0.0 vol.%)			(0.019 vol.%)			(0.046 vol.%)		
		Thickness (μm)			Thickness (μm)			Thickness (μm)
#1		17.68	#1		18.04	#1		18.67
#2		18.13	#2		19.65	#2		17.07
#3		18.59	#3		17.41	#3		19.71
#4		19.53	#4		19.14	#4		18.98
#5		18.62	#5		17.93	#5		18.44
#6		19.31	#6		17.92	#6		18.80
Average		18.64 ± 0.70	Average		18.35 ± 0.85	Average		18.61 ± 0.87
GPL-50			GPL-75			GPL-100		
(0.098 vol.%)			(0.142 vol.%)			(0.190 vol.%)		
		Thickness (μm)			Thickness (μm)			Thickness (μm)
#1		19.89	#1		19.59	#1		18.08
#2		19.10	#2		18.38	#2		19.36
#3		17.35	#3		19.57	#3		17.59
#4		18.31	#4		19.52	#4		17.89
#5		18.91	#5		18.71	#5		18.62
#6		18.77	#6		17.91	#6		18.44
Average		18.72 ± 0.84	Average		18.95 ± 0.72	Average		18.32 ± 0.63

(2) As mentioned earlier (please see the detailed answer for the Q2), we adjusted the thickness of the 1-layer GPM to manufacture a desired GPL. To clarify it, we revised manuscript as follows.

→ In page 8,

The volume fraction (V_G) and the interval of the graphene fillers can be simply adjusted by the concentration of the PMMA and the spin coating speed (Supplementary Fig. 4).

→ In page 15,

... Then, PMMA (950k, Microchem) dissolved in anisole was coated on the as-grown graphene, and satisfied sufficient thickness for floating with different concentrations of PMMA solution (4, 8, and 12 wt.%) and spinning speeds (500, 1,000, 1,500, 2,500, and 4,000) rpm to modulate the graphene volume fraction (Supporting information Fig. 4). ...

Q5. The mechanical properties in Fig. 2 were measured through using samples with similar total thicknesses. This means that the thickness of embedded PMMA layers in GPLs is different in these experiments. How about the mechanical properties of GPLs with same PMMA layer thickness but different total thickness?

→ We appreciate reviewer's helpful and valuable comment.

Following the reviewer's suggestion, we prepared GPL with 10-, 50-, 100-layer by stacking of the same GPM (PMMA A4, 1500 rpm). The thickness of 10-layer GPL, 50-layer GPL, and 100-layer GPL was 1.74 μm , 9.68 μm , and 17.90 μm , respectively (Figure R1a-c).

The average fracture forces (F) of the 3 samples obtained from the tensile tests was 21.64 ± 1.59 cN, 109.84 ± 1.96 cN, and 196.24 ± 7.72 cN, respectively, as the number of laminated layers increased (Figure R1d-f). The average strength and modulus of the 3 samples were calculated by the stress equation ($\sigma = F/A$) (Figure R1g).

Average strength: 86.57 ± 6.36 MPa for the 10-layer GPL, 141.83 ± 2.53 MPa for the 50-layer GPL, and 135.34 ± 5.33 MPa for the 100-layer GPL

Average modulus: 5.75 ± 0.11 GPa for the 10-layer GPL, 5.30 ± 0.19 GPa for the 50-layer GPL, and 5.32 ± 0.32 GPa for the 100-layer GPL

Except for the strength of the 10-layer GPL, we found that these average mechanical properties were all to be within the error range of the GPL-100 in Fig. 2. Therefore, we could conclude that the GPLs of the same graphene volume fraction have similar mechanical properties.

The 10-layer GPL was fractured even with small deformation. This behavior of the 10-layer GPL was observed in multiple tensile tests. After stress-strain test, we found that the fracture was raised far from the middle of the samples (Inset of Figure R1d). Thus, we believe that in the process of cutting the samples of ultra-thin films for specimen fabrication, undesired cracks could be formed at the edge of the film, which resulted in limitations in the measurements of the mechanical properties [*Nat. Commun.* **4**, 2520, (2013); *Acta Mater.* **48**, 2843-2857 (2000)].

→ A data for Response Letter,

Fig. R1 | Mechanical characterization of GPLs stacked by the 10-, 50-, and 100-layer of the GPM (A4, 1500 rpm). a-c. Cross sectional SEM image of 10-, 50-, and 100-layer GPL, respectively. The concentration of the PMMA solution and spin coating speed for the GPM are A4 and 1500 rpm, respectively. The 10-, 50-, and 100-layer GPL have a thickness of 1.74 μm , 9.68 μm , and 17.90 μm , respectively. Scale bar is 10 μm . d-f. Force versus strain curve and g. Stress versus strain curve of 10-, 50-, and 100-layer GPL. Inset in (d) shows a photograph of 10-layer GPL after the fracture. Scale bar is 0.5 cm.

Q6. The authors use the number of layers to characterize the mechanical property. However, the volume content of graphene is the direct factor for the mechanical performance. The volume content of graphene should be given in the Fig. 2c and d.

→ We thank again the reviewer's detailed comments. We added the graphene volume fraction in main-text (including Fig. 2) and Supplementary Information.

Q7. The authors analyze the mechanical properties of S-GPL, T_g -GPL, and GPL, and indicated that the mechanical performance of GPL was largely enhanced. The reviewer thinks this enhancement mainly caused by the hot-rolling mill process which resulting in a compact interface between PMMA and graphene; the reinforcement efficiency of float stacking process may have a positive effect but not the main factor. My question is how about the mechanical performance of multilayered graphene-PMMA laminate fabricated by simply stacking and hot-pressing. The simple and easy manufacturing method is significant for engineering practices.

→ We thank the reviewer's critical and relevant comments.

In order to understand the effect of hot-rolling mill process, we fabricated wet-transferred graphene-PMMA laminate with 25 layers of GPM (WT-GPL-25, PMMA A4, 500 rpm) and performed the tensile test before and after hot-rolling mill process. We confirm that hot-rolling mill process help to improve the fracture strength from 60.89 to 77.92 MPa, and the modulus from 3.73 to 4.74 GPa, respectively. However, the overall mechanical performance of the WT-GPL-25 was lower than that of T_g -GPL with 25 layers of GPM (PMMA A4, 500 rpm). Noted that mechanical properties of S-GPL, T_g -GPL, and GPL with 25 layers of GPM (PMMA A4, 500 rpm) are described in Fig. 3.

Fig. R2 | Mechanical characterization of wet-transferred graphene-PMMA laminate with 25 layers of GPM (WT-GPL-25) before and after hot-rolling mill process. Stress-strain response of WT-GPL-25 before (black) and after (red) hot-rolling mill process. After hot rolling mill process, the fracture strength increased from 60.89 to 77.92 MPa and the modulus from 3.73 to 4.74 GPa, respectively.

The inferior characteristics of the wet-transferred graphene-PMMA laminate stem from the method itself. During the wet-assisted transfer process, the presence of water at the surface of film inevitably leads to the formation of blisters and bubbles at the interface of the interlayer of

composite [Ref. *Adv. Funct. Mater.* **26**, 2070 (2016); *Proc. R. Soc. A*, **324**, 301–313 (1971)]. We fully agree with the reviewer's comment that the hot-rolling mill process provides a compact interface and contributes to the enhancement of mechanical properties. However, even if a strong compressive stress is applied to the laminate by the hot-rolling mill process, the water-trap could not be completely removed and remains irreversible folds, wrinkles, and cracks. On the contrary, the overall float stacking method provides a conformal and dry contact between each layer by rolling at above the T_g . Web-tension caused by capillarity action also provides continuous contact along the surface of the roll, suppressing the generation of wrinkles and folds. Thus, we were able to reduce the generation of defects mainly caused by water-trap.

In this study, as the reviewer pointed out, hot-rolling mills process act as one important key process. However, we strongly believe that interface optimization in the lamination process could be a crucial prerequisite to maximize reinforcement efficiency, and our float-stacking approach provides it.

II. Response to comments of Reviewer #2

This paper describes a new method that uses the free surface of a liquid as a support for the production of composites alternating sheets of PMMA with sheets of polycrystalline graphene produced by CVD. The topic is innovative and treated with the necessary experimental scientific rigor. There are also two mechanical and thermal analytical analyses and one thermal numerical analysis. The strength of the paper is certainly the experimental part. However, the mechanical analytical approach is important as it suggests how the composite is done properly, returning a Young's modulus equal to the theoretical graphene modulus of about 1 TPa and a strength of about 1/3 of the theoretical value of 100 GPa but compatible with the expected values of polycrystalline graphene obtained via CVD.

Q1. I would add in the main text in the mechanics section the thicknesses of PMMA and/or the whole specimen i.e. all the data entered in the rule of mixture.

→ We apologize for the omission and thank the reviewer for pointing this out. We added PMMA thickness data from all GPLs entered in the rule of mixture and revised our manuscript as follows.

→ In page 8,

The volume fraction (V_G) and the interval of the graphene fillers can be simply adjusted by the concentration of the PMMA and the spin coating speed (Supplementary Fig. 4). ... We prepared the GPLs in which the graphene fillers of (0, 10, 25, 50, 75, and 100) layers, referred to as GPL-0, GPL-10, GPL-25, GPL-50, GPL-75, and GPL-100, respectively, were aligned (Fig. 2a). For the tensile stress test, the GPL samples were prepared with a length of 3 cm, and they all had similar (or almost the same) thicknesses (Supplementary Fig. 5 and Supplementary Table 1).

→ In page 10,

... To explore the remarkable reinforcing factors in our GPL samples, as compared with previous results of graphene/PMMA composites, we prepared three different samples with 25 layers of GPM (A4, 500 rpm), the simply-stacked GPL (S-GPL), stacked above the T_g (T_g -GPL), and GPL, respectively (Supplementary Fig. 7), and then performed a tensile test. The volume fraction of GPL with 25 layers of GPM (A4, 500 rpm) is 0.098%. ...

→ In page 15,

... Then, PMMA (950k, Microchem) dissolved in anisole was coated on the as-grown graphene, and satisfied sufficient thickness for floating with different concentrations of

PMMA solution (4, 8, and 12 wt.%) and spinning speeds (500, 1,000, 1,500, 2,500, and 4,000) rpm to modulate the graphene volume fraction (Supporting information Fig. 4). ...

Supplementary Fig. 4 | Thickness analysis of the 1 layer (L)-GPM according to the concentration of PMMA solution and coating speed. After spin coating the PMMA on the graphene, the GPMs transferred onto the 300 nm SiO₂/Si. Thickness of the GPMs was measured by the Alpha-step. To manufacture the GPLs of the same thickness, we used different GPMs under the following conditions: GPL-10 (PMMA A12, 1500 rpm), GPL-25 (PMMA A8, 1500 rpm), GPL-50 (PMMA A4, 500 rpm), GPL-75 (PMMA A4, 1000 rpm), GPL-100 (PMMA A4, 1500 rpm). The graphene volume fractions for GPL-10, GPL-25, GPL-50, GPL-75, and GPL-100 are 0.019 vol.%, 0.046 vol.%, 0.098 vol.%, 0.142 vol.%, and 0.190 vol.%, respectively.

Fig. 2 | Mechanical characterization of the GPLs. **a**, Photograph of the GPLs with different number of graphene reinforcement layers of (0, 10, 25, 50, 75, and 100). **b**, Typical stress–strain responses of GPLs. Inset shows photograph of GPL–100 before and after tensile test. **c** and **d**, Fracture strength, and Young’s modulus, of the GPLs. Both mechanical properties of GPLs linearly increased according to the number (or the volume fraction) of embedded graphene layers. **e**, Comparison of the specific strength of GPLs with that of the previously reported graphene/PMMA composites.

Q2. I would avoid writing that capillarity forces are able to remove all defects as initially reported in the paper as unlikely, and as moreover admitted in the second part of the paper.

→ We entirely agree with the reviewer's concern the capillarity forces could not completely remove the defects during the float stacking process. We described in the manuscript that the density of defects (voids, wrinkles, and folds) could be reduced through web-tension in the float-stacking process. However, we have described in the manuscript that the remaining small voids were completely removed through the hot-rolling mill process and agree that our expression is overstated. Thus, we revised the manuscript as follows.

→ In page 7,

... Lastly, we **nearly** eliminated the interlayer voids and bubbles presented at interlayers by squeezing the stacked–GPM through a hot-rolling mill process (the effect and detailed procedures of heat-treatment are discussed in the following text). ...

→ In page 11,

However, by rolling up above the T_g temperature, the viscosity of the highly elastic PMMA film was increased, inducing conformal contact between each layer (Supplementary Fig. 9). Thereafter, the small voids present in the T_g -GPL were **effectively** removed through the hot-rolling mill process (Figs. 3b, c and Supplementary Fig. 10).

→ In order to clarify the effect of hot-rolling mill process, we further performed optical microscope analysis (both bright field (BF) and dark field (DF) mode), and AFM analysis to precisely verify the morphology of the GPL sample. Note that wet-transferred graphene-PMMA laminate (WT-GPL) was prepared for comparison. Both samples (GPL and WT-GPL) were stacked with 10 layers of GPM (PMMA A4, 1500 rpm). In contrast to the WT-GPL, where various defects (wrinkles, bubbles, holes, and water traps) were observed, no physical defects were found on the surface and interlayers of GPL (Supplementary Fig. 10). The root mean square (RMS) roughness of the GPL was 69.26 nm, which is nearly half that of the WT-GPL.

We believe that these experimental results strongly supported that the float-stacking process could effectively eliminate interlayer defects, allowing us strong interfacial adhesion of laminate. We added the OM and AFM data to the Supplementary Information.

→ In Supplementary Information,

Supplementary Fig. 10 | Morphology analysis of GPL and wet-transferred graphene-PMMA laminate (WT-GPL) with 10 layers of GPM (PMMA A4, 1500 rpm). High magnification (Left) bright field and (Right) dark field OM images of GPL

and WT-GPL. In the corresponding AFM images, the root mean square of GPL is 69.26 nm and WT-GPL is 138.40 nm. Scale bar is 50 and 20 μm , respectively.

Q3. The length of the specimens is 1 cm and could lead to inherently higher strength measurements given the smaller size (and relative lower likelihood of finding defects) than the traditional ones at which composites are tested (closer to 10 cm than 1).

→ We thanks to the reviewer's critical and relevant comments.

We analyzed the mechanical properties of the relatively small-sized GPL specimens to precisely observe the mechanical behavior of embedding graphene layers. Then, we utilized a high-resolution load cell of 210 cN (resolution within 0.0001 cN). As a result, we observed an ideal increase in strength of GPLs according to the number of graphene layers.

→ We agree with the reviewer's comment that the likelihood of detect randomly located defects is lower in small specimen sizes. Accordingly, we manufactured a large-sized GPL-100 of $4.3 \times 9.8 \text{ cm}^2$, and performed a tensile test with a 9 cm length. The strain-stress curve in Figure. R3 shows statistical tendency for the mechanical properties of the large-scale GPL. The average strength and modulus were calculated to be $131.76 \pm 8.44 \text{ MPa}$ and $5.36 \pm 0.28 \text{ GPa}$, respectively. We confirmed that mechanical properties of the both samples (GPL-100 and large-sized GPL-100) were overlapped within the error range. Therefore, we believe that the mechanical properties (and density of defects) of GPLs manufactured by float-stacking strategy are uniform.

→ A data for response letter,

Fig. R3 | Mechanical characterization of large-sized GPL-100. The average strength and modulus were calculated to be $131.76 \pm 8.44 \text{ MPa}$ and $5.36 \pm 0.28 \text{ GPa}$. The specimens were all

measured with a length of 9 cm and width of 4.3 cm. The large-size GPL-100 was fabricated from a roll with a diameter of 3.2 cm, and all processes were the identical to the typical GPL-100 manufacturing process.

→ We also added photograph of the large-sized GPL in Fig. 1 and revised the manuscripts as follows.

→ In page 5-6,

Figure 1b shows a photo image of the $18.32 \pm 0.63 \mu\text{m}$ thick, $9.8 \text{ cm} \times 4.3 \text{ cm}$ GPL, in which 100 layers of GPM are uniformly stacked.

Fig. 1 | Manufacture of the GPL via float-stacking process. **a**, Schematic of the float-stacking process of the GPL. (i) Floating the GPM on DI water bath after wet-etching of the bottom Cu foil, (ii) Layer-by-layer stacking of GPMs by rolling process, (iii) Cutting and unfolding of stacked-GPM, and (iv) Hot-rolling mill process of the stacked-GPM. **b**, Photograph of the as-prepared GPL with 100 layers of GPM (GPL-100). **c and d**, Cross-sectional SEM and TEM images of the GPL-100. Monolayer graphene is placed between the PMMA matrix without structure defects. Scale bar: $100 \mu\text{m}$ and 5 nm , respectively.

Q4. I think the paper is much improvable in the analytical and numerical parts, but since it is primarily experimental it can be published nearly in this form in Nature Communications. Almost every time I see Pa writted with p and it should be corrected.

→ We apologies for our mistakes and thank the reviewer for pointing this out. We carefully checked our manuscript again and corrected all the typos.

Q5. Using a law derived from atomistic simulations from the grain size reported in the paper I find a graphene resistance of about 80 and not 33 GPa as calculated in this paper, but there could be many reasons for this difference.

→ We thank the reviewer's critical comment. We estimated the strength of graphene (33 GPa) in the GPL through rule of mixture and we believe that this value falls within the range of the strength of a typical polycrystalline CVD-graphene.

However, several studies have reported high strength values for graphene, which are similar to that of perfect graphene. Therefore, we carefully investigated previously published studies on the mechanical properties of graphene once again (Table R2) and we found that differences in strength could be attributed to the following reasons:

Table R2. Compilation of strength and Young's modulus of various graphene.

Materials	Method	Type of graphene	Strength (GPa)	Modulus (GPa)	Ref
Graphene (Mechanical exfoliation)	AFM nanoindentation	Single crystalline	130 ± 10	1015 ± 90	Science , 321 , 385–388, (2008)
		Single crystalline	125	1039	Nat. Commun. 8 , 15815 (2017)
Graphene (CVD)	AFM nanoindentation	Bi-crystalline	90-99	1012	Science 340 , 1073-1076 (2013)
		Bi-crystalline (low-tilt angles)	39-78	-	
		Single crystalline	45 ± 10	1000	Phys. Status Solidi RRL 9 , 564–569, (2015)
		Poly crystalline	16.4 ± 5.1	1000	
		Single crystalline	90-94	-	Nat. Commun. 4 , 2811 (2013)
		Bi-crystalline	83	-	
		Bi-crystalline (low-tilt angles)	53	-	
	Poly crystalline	35	-	Nano Lett. 11 , 2259–2263 (2011)	
	SEM tensile testing	Single crystalline	50-60	922	Nat. Commun. 11 , 284 (2020)
	Tensile testing	Single crystalline	3.7–4.5	683-908	Adv. Mater. 30 , 1800888, (2018)
		Poly crystalline	3.33	637-793	
Micro tensile testing	Poly crystalline (100 layer)	5.8	290	Adv. Mater. 31 , 1903039, (2019)	
Macro tensile testing	Poly crystalline (100 layer)	0.7	62		
Graphene (Theoretical)	DFT calculation	Single crystalline (armchair)	121	1050	Phys. Rev. B 76 , 064120 (2007)
		Single crystalline (zigzag)	110	1050	
		Bi-crystalline (tilt angle : 0 °)	93	-	ACS nano 6 , 2704-2711 (2012)
		Bi-crystalline (tilt angle : 42 °)	46	-	
	Molecular simulations	Single crystalline (armchair)	109	1010 ± 30	Nano lett. 9 , 3012-3015 (2009)
		Single crystalline (zigzag)	90	1010 ± 30	
		Bi-crystalline (tilt angle : 27.8 °)	100	-	Nat. Mater. 11 ,759–763 (2012)
		Bi-crystalline (tilt angle : 5.7 °)	33	-	
Nano crystalline	35-60	558 ± 2	Phys. Rev. B 85 , 195447 (2012)		

→ (i) Nanoindentation of micron-sized suspended graphene films using AFM is well known approach to measure the mechanical properties of single-atom-thick graphene [Ref. *Science* **321**, 385–388 (2008); *Phys. Status Solidi RRL* **9**, 564–569 (2015); *Nat. Commun.* **8**, 15815 (2017)]. They recorded the superior modulus and strength of various types of graphene. In addition, failure strengths of 50-60 GPa have been reported for freestanding graphene films using In situ SEM tensile testing [Ref. *Nat. Commun.* **11**, 284 (2020)]. However, the strength of centimeter-scale CVD graphene can significantly differ from results achieved with micro-sized graphene flakes. For example, a fracture strength of 3.33 GPa was calculated from a uniaxial tensile test of camphor-assisted transferred graphene, and a macro-scale test of mixed stacking ordered 100-layer CVD-graphene revealed a strength eight times lower than the micro-tested [Ref. *Adv. Mater.* **30**, 1800888 (2018); *Adv. Mater.* **31**, 1903039 (2019)]. The presence of rare discontinuous regions and inevitable thermodynamic defects in CVD graphene are responsible for strength degradation in macroscopic graphene. In particular, cracks propagate from nano-sized defects present in the graphene, leading to brittle fracture [Ref. *Nat. Commun.* **5**, 3752 (2014); *Nat. Commun.* **3**, 28 (2022)]. In contrast to the theoretical fracture strain of graphene (~20%), experimental studies of centimeter-scale CVD graphene-based lamination composites have further confirmed a limited fracture strain trend due to the imperfections in graphene [Ref. *Nat. Commun.* **12**, 4655 (2021); *ACS Appl. Mater. Interfaces* **7**, 10702–10709 (2015); *Adv. Mater.* **30**, 1707449 (2018)].

→ (ii) In our study, the average grain size of Cu-grown CVD graphene was calculated to be $14.18 \pm 3.41 \mu\text{m}$. As mentioned by the reviewer, several research reported higher graphene resistance over our results despite the larger or smaller grain size [Ref. *Science*, **321**, 385–388 (2008); *Nat. Commun.*, **4**, 2811 (2013); *Carbon* **72**, 200-206 (2014)]. The grain size-dependent strength reduction caused by grain boundaries in graphene is evident. However, the fracture mechanism of polycrystalline graphene is complex and can be influenced by various factors. For example, the fracture strength can vary depending on the polarity, hydrogenation, chirality and chiral angle of the grain boundary [Ref. *Comput. Mater. Sci.* **91**, 56-61 (2014); *J. Phys. Chem. C* **118**, 13769–13774 (2014); *J. Mech. Phys. Solids.* **110**, 118-136 (2018)]. In particular, several groups have reported strength reductions of 40~70% depending on grain boundary misorientation angles in bi-crystal graphene systems [Ref. *Carbon* **50**, 3465-3472 (2012); *Nat. Commun.* **4**, 2811 (2013); *Science* **340**, 1073-1076 (2013)]. Despite the large-sized grains, the GPL embedded with 100 layers of graphene contains numerous grain boundaries. Since cracks initiate and propagate at weak locations along grain boundaries or small defects, practical applications of polycrystalline graphene films are likely to belong to the lower strength range.

→ (iii) Many research groups have confirmed that the strength in CVD graphene measured by experimental is lower than the theoretical value [Ref. *Nature* **469**, 389–393 (2011); *Nano Lett.* **11**,

2259–2263 (2011); *ACS Nano* **8**, 10246–10251 (2014)]. In addition, Suk et al. reported strengths of 30–60 GPa (average 45.4 GPa) in single crystal graphene tested by in-situ indentation, while for polycrystalline graphene, and a lower average of 16.4 GPa for polycrystalline graphene [Ref. *Phys. Status Solidi* **9**, 564–569 (2015)]. Contrary to the pristine graphene expected in atomistic simulations, as we mentioned before, the inherent defects in the CVD process led to the rapid fracture of graphene. The possible defects that can arise during the sample preparation or testing of CVD graphene can also introduce variations in strength and contribute to the differences between theoretical and experimental mechanical behavior [Ref. *Adv. Mater.* **30**, 1800888 (2018)].

REVIEWER COMMENTS

Reviewer #1 (Remarks to the Author):

This manuscript has described a method that essentially utilizes liquid-air interface to fabricate laminated composites with PMMA sheets and CVD graphene. It sort of addresses the comments mentioned by other Reviewers. However, it still lacks further elaboration and analysis on the mechanical reinforcement. Therefore, I would suggest a major revision before publication.

1. Similar to the Reviewer #1 that I thought this manuscript has some overlap with the work published by Vlassiuk [ACS Appl. Mater. Interfaces 7, 10702-10709 (2015).] and Liu [Science 353, 364-367 (2016).], though there is some difference from the above two works. The Authors should further highlight the novelty of the manuscript.

2. The superior mechanical properties are one of key point of this manuscript but explanation might be oversimplified, particularly on the strength. I am not entirely sure how appropriate to use rule of mixture to predict strength (though used in many literature), as the strength is actually limited by the lower limit of the components, not as a weighted sum of all, like what modulus does. The Authors are suggested to add more analysis to interpret the superior mechanical reinforcement (could be either micromechanics experiment or simulation, or others).

3. Is there any experiment about the modulus of the CVD graphene? As is known, CVD graphene has large amount of grain boundaries and defects, which may cause its properties below what a single crystal graphene has. There could be other mechanisms on the strength side as the strength (Table R1) is even slightly higher than that of single crystal graphene. The Authors should show the similarity and difference between the CVD graphene used and single crystal graphene, on both modulus and strength.

Reviewer #3 (Remarks to the Author):

I review many papers and I found this one to be one of the most enjoyable I have read for a long time. The authors have produced a near-perfect model nanocomposite and characterized its mechanical and thermal properties with a high degree of precision.

I note that an earlier version of the paper was assessed by two reviewers - perhaps for a higher impact Journal in the same family? Their main criticism was over its novelty compared particularly with refs 10 and 12. The authors seem to have addressed their points well making the paper now worthy of publication in Nature Communications. Overall it seems that the material produced in this present study is of superior quality making it now possible to use the simple rule of mixtures to evaluate fundamental properties.

The main selling point of the work to me is the multi-functional aspect in that they can obtain both fundamental mechanical and thermal parameters from the same material. The previous reviewers did not seem to comment on the thermal aspects.

I do, however, have just one minor issue to raise. On line 202 (and in the SI) they say that the viscosity was increased by raising the temperature above T_g . Surely the viscosity of a polymer decreases as the temperature is increases? I think they mean to say that conformal contact would be induced by a lower viscosity.

Authors' Reply to the Reviewers' Comments on:

Float-stacked multi-functional graphene–PMMA laminate

By Seung-Il Kim, Ji-Yun Moon, Seok-Ki Hyeong, Soheil Ghods, Jin-Su Kim, Jun-Hui Choi, Dong Seop Park, Sukang Bae, Sung Ho Cho, Seoung-Ki Lee, and Jae-Hyun Lee

We deeply appreciate the effort that the editor and reviewers have taken in reviewing our manuscript. Here we include the answer to the referees and the revised manuscript. Changes have been carried out according to the comments and highlighted as red color in the revised version of the manuscript. We hope that our reply properly addressed the points of the comments.

I. Response to comments of Reviewer #1

This manuscript has described a method that essentially utilizes liquid-air interface to fabricate laminated composites with PMMA sheets and CVD graphene. It sort of addresses the comments mentioned by other Reviewers. However, it still lacks further elaboration and analysis on the mechanical reinforcement. Therefore, I would suggest a major revision before publication. listed below.

Q1. Similar to the Reviewer #1 that I thought this manuscript has some overlap with the work published by Vlassiounk [ACS Appl. Mater. Interfaces 7, 10702-10709 (2015).] and Liu [Science 353, 364-367 (2016).], though there is some difference from the above two works. The Authors should further highlight the novelty of the manuscript.

→ We thank the reviewer for taking the time and effort necessary to carefully review our manuscript. We believe that our float stacking strategy provide a breakthrough to overcome limitations in the field of manufacturing of multifunctional nanocomposites. As the reviewer pointed out, we further highlighted the novelty and differences of our results in introduction part as follows.

→ In page 3-4,

... Herein, we presented a scalable manufacturing approach, a float-stacking strategy, for a multi-functional graphene-PMMA laminates (GPLs), in which highly crystalline semi-infinite graphene fillers are uniformly aligned in the polymer matrix. To quickly and precisely insert monolayer graphene into the PMMA matrix without mechanical destruction, we floated the delicate and thin graphene–PMMA membrane (GPM) that

only relies on water support on a water–air interface, and wound it up layer-by-layer. During the stacking process, the inherent water meniscus at the interface between the GPM and deionized (DI) water continuously induced web tension of the GPM, suppressing the structural defects (e.g., wrinkles and folding). Moreover, by the rolling-up and hot-rolling mill process above the glass transition temperature (T_g), the viscosity of the highly elastic PMMA film was increased, inducing dry and conformal contact between each GPM layer as well as pre-tension of the GPLs...

Q2. The superior mechanical properties are one of key point of this manuscript but explanation might be oversimplified, particularly on the strength. I am not entirely sure how appropriate to use rule of mixture to predict strength (though used in many literature), as the strength is actually limited by the lower limit of the components, not as a weighted sum of all, like what modulus does. The Authors are suggested to add more analysis to interpret the superior mechanical reinforcement (could be either micromechanics experiment or simulation, or others).

→ We thank the reviewer's critical comments. Following the reviewer's suggestion, we designed a material model and conducted tensile simulations to clarify the mechanical behavior of GPLs.

The strength of the structure was determined through a finite element method (FEM) analysis using COMSOL Multiphysics. This analysis employed the linear elastic relation provided in the structural mechanics module in COMSOL. Supplementary Fig. 6a depicts the strain diagram for various graphene-PMMA structures corresponding to different numbers of graphene layers (0, 10, 25, 50, 75, and 100 layers) under a stress level of 77.9 MPa. Notably, an observed trend reveals that with an increasing number of graphene layers, the strain in the structure decreases. This reduction in strain at a specific stress level signifies an enhancement in the tensile strength of the structure. Also, the experimental stress-strain results of GPL-0, GPL-50, and GPL-100 (blue lines) are in agreement with the simulation results (dotted line) from the initial strain, verifying the increasing strength with the number of aligned graphene layers in the composite (Supplementary Figs. 6b-d). Note that the strength of the GPL may be slightly lower than that of the simulation results due to the occurrence of fractures within the structure. Consequently, the tensile simulation validates the reinforcement achieved with graphene fillers, supporting our mechanical performance.

We believe that the reviewer's valuable suggestions have been very helpful to strengthening the reliability of our mechanical strength results and improve the overall manuscript. Detailed information on simulation was added in Supplementary Information and revised manuscripts as follows.

→ In page 8-9,

... The Young's modulus of the GPL was enhanced by 261.26 % from ((3.33 ± 0.15) to (5.37 ± 0.31)) GPa (Fig. 2d). The Young's modulus of GPL-10 to GPL-75 was ((3.40 ± 0.17), (3.92 ± 0.22), (4.24 ± 0.10), and (4.70 ± 0.19)) GPa, respectively (see Supplementary Table 2). To clarify the mechanical behavior of GPL, finite element method (FEM) simulations were employed to model graphene-PMMA structures corresponding to different numbers of graphene layers (0, 10, 25, 50, 75, and 100 layers) (Supplementary Fig. 6). An inverse relationship trend has been observed, reveal that as the number of graphene layers increases in the composite, the strain in the structure decreases. Based on simulation results, a reduction in strain at a specific stress level signifies an enhancement in the tensile strength of the structure. Moreover, the experimental stress-strain results agree with the simulation results, verifying the mechanical properties of GPL.

→ In Methods,

FEM Simulation. The strengths of distinct structures with varying numbers of graphene layers are evaluated through FEM analysis employing COMSOL Multiphysics. For uniaxial tension testing of graphene-PMMA structure, we directly modeled the geometry corresponding to GPL-0 to 100 and employed the linear elastic relationships provided in the mechanics section for accurate estimation. In this simulation, a fixed boundary was established, and an applied force was introduced to the free boundary, and the resulting deformation, quantified as static displacement, was measured. Tensile simulation was computed according to the following relation as $-\vec{E}\nabla \cdot \mathbf{u} = F_v/A$, where \vec{E} represents the elastic constant, A denotes the force application area ($W \times t$: 0.6 mm × 18 μm), \mathbf{u} represents displacement, F_v represents the applied force inducing strain.

→ In Supplementary Information,

Supplementary Fig. 6 | FEM tensile simulations and stress–strain responses. **a.** Strain diagrams of graphene-PMMA structures corresponding to different graphene layer counts (0, 10, 25, 50, 75, 100 layers) at a stress level of 77.9 MPa. A distinct reduction in strain was observed as the number of graphene layers increases. **b-d.** Comparison between the experimental (blue lines) and simulated (dotted lines) stress-strain curves of GPL-0, GPL-50, and GPL-100.

Q3. Is there any experiment about the modulus of the CVD graphene? As is known, CVD graphene has large amount of grain boundaries and defects, which may cause its properties below what a single crystal graphene has. There could be other mechanisms on the strength side as the strength (Table R1) is even slightly higher than that of single crystal graphene. The Authors should show the similarity and difference between the CVD graphene used and single crystal graphene, on both modulus and strength.

→ We appreciate the reviewer’s helpful comments. As suggested by the reviewer, we carefully compared the mechanical properties of single crystal graphene, and our graphene as follows.

(1) In light of the persistent linear correlation between graphene volume fraction and modulus was followed in all GPL with a precisely aligned graphene filler from 0 to 100 layers, leveraging the rule of mixture, we reliably derived the modulus of polycrystalline CVD graphene as 1.09 TPa. As mentioned by the reviewer, the modulus of polycrystalline CVD graphene aligned within the GPL is closing to that of single-crystal graphene (1.0 ± 0.1 TPa), signifies the presence of

different mechanism beyond the alignment of graphene for reinforcing the stiffness of nanocomposite [Ref. *Science* **321**, 385-388 (2008)].

→ A prominent distinction between single crystal graphene and polycrystalline CVD-graphene is clearly the presence or absence of grain boundaries. However, several studies have shown that the modulus of smaller polycrystalline graphene are not statistically different from that of single crystal graphene; are consistent with theoretical predictions in the absence of grain boundaries [Ref. *Science* **340**, 1073-1076 (2013); *Nat. Commun.* **4**, 2811 (2013)]. Similarly, a series studies with larger-scale polycrystalline CVD graphene membranes, the Young's modulus is within a close to that of single-crystal graphene, but a wider distribution has been observed [Ref. *Nat. Commun.* **11**, 284 (2020); *Chem. Mater.* **32**, 6078-6084 (2020)]. The error in this wide distribution is attributed to the presence of wrinkles in the CVD-grown film. The inevitable wrinkles that are produced in the process of graphene growth induce out-of-plane corrugations in the graphene membrane, softening its stiffness [Ref. *Nano Lett.* **11**, 2259-2263 (2011)].

→ As mentioned earlier, we note that the graphene in the float-stacked GPL is pre-tensioned. The uniaxial strain induced by the heat treatment stretches the graphene, which has discussed in detail at the Supplementary note. Stretching of wrinkles in the graphene membrane improve the stiffness of the softened graphene, leads to high modulus of the nanocomposite with additional reinforcement effect. Overall, polycrystalline CVD graphene in laminate feature a lower modulus with respect to single-crystal graphene because of the defects, wrinkles and folding that originate in its growth. We controlled the pre-tension of the aligned graphene to achieve a Young's modulus close to the pristine graphene, deriving to reinforcement of the mechanical performance of the composite.

(2) Graphene, mono-atomic layer of graphite, with carbon atoms arranged in a hexagonal lattice of sp^2 bonds that exhibit superior mechanical properties; single crystalline pristine graphene possesses an intrinsic tensile strength of 130 GPa [Ref. *Science* **321**, 385-388 (2008)]. However, in practical graphene synthesized from CVD, which is a promising technology for large-scale production, is predominately polycrystalline and always imperfections caused by the presence of carbon structures that deviate from the standard honeycomb lattice of graphene. Numerous studies have shown that the fracture strength of graphene is highly dependent on the presence of several types of defects (e.g., voids, nano-cracks, and chemical impurities), including grain boundaries [Ref. *Science* **330**, 946-948 (2010); *Nature* **469**, 389-392 (2011); *Nat. Commun.* **11**, 284 (2020)]. As a result, fracture is one of the most conspicuous issues for the mechanical behavior of polycrystalline CVD graphene. Experimental studies reveal that the fracture strength of

polycrystalline graphene is ~35 GPa, a quarter of that of single-crystal graphene, agreeing with our graphene from the GPL [Ref. *Nature* **469**, 389-392 (2011); *Nano Lett.* **11**, 2259-2263 (2011)].

II. Response to comments of Reviewer #3

I review many papers and I found this one to be one of the most enjoyable I have read for a long time. The authors have produced a near-perfect model nanocomposite and characterized its mechanical and thermal properties with a high degree of precision.

*I note that an earlier version of the paper was assessed by two reviewers - perhaps for a higher impact Journal in the same family? Their main criticism was over its novelty compared particularly with refs 10 and 12. The authors seem to have addressed their points well making the paper now worthy of publication in *Nature Communications*. Overall it seems that the material produced in this present study is of superior quality making it now possible to use the simple rule of mixtures to evaluate fundamental properties.*

The main selling point of the work to me is the multi-functional aspect in that they can obtain both fundamental mechanical and thermal parameters from the same material. The previous reviewers did not seem to comment on the thermal aspects.

Q1. I do, however, have just one minor issue to raise. On line 202 (and in the SI) they say that the viscosity was increased by raising the temperature above T_g . Surely the viscosity of a polymer decreases as the temperature is increases? I think they mean to say that conformal contact would be induced by a lower viscosity.

→ We apologize for the confusion and thank the reviewer for pointing this out. We modified manuscript as follows

→ In page 11,

... Thus, the wrinkles and hollow regions that lead to interlayer sliding were frequently observed on the surface and interlayer of the S-GPL, respectively³⁶. **However, rolling above the T_g temperature increased the viscosity of the highly elastic PMMA film, inducing conformal contact between each layer (Supplementary Fig. 10)³⁷.**

[37] *Nat. Commun.* **13**, 4409 (2022).

REVIEWERS' COMMENTS

Reviewer #1 (Remarks to the Author):

The Authors have addressed my comments so the manuscript can be accepted.

Authors' Reply to the Reviewers' Comments on:

Float-stacked graphene–PMMA laminate

By Seung-Il Kim, Ji-Yun Moon, Seok-Ki Hyeong, Soheil Ghods, Jin-Su Kim, Jun-Hui Choi, Dong Seop Park, Sukang Bae, Sung Ho Cho, Seoung-Ki Lee, and Jae-Hyun Lee

We deeply appreciate the effort that the editor and reviewers have taken in reviewing our manuscript. Here we include the answer to the referees and the revised manuscript.

I. Response to comments of Reviewer #1

The Authors have addressed my comments so the manuscript can be accepted.

→ We appreciate the reviewer for the positive comments.